# Intracellular Localization in Zebrafish Muscle and Conserved Sequence Features Suggest Roles for Gelatinase A Moonlighting in Sarcomere Maintenance

**DOI:** 10.3390/biomedicines7040093

**Published:** 2019-11-29

**Authors:** Amina M. Fallata, Rachael A. Wyatt, Julie M. Levesque, Antoine Dufour, Christopher M. Overall, Bryan D. Crawford

**Affiliations:** 1Department of Biology, University of New Brunswick, Fredericton, NB E3B 5A3, Canada; afallata-404@hotmail.com (A.M.F.); r.a.wyatt@unb.ca (R.A.W.); jl395146@gmail.com (J.M.L.); 2Department of Physiology & Pharmacology, University of Calgary, Calgary, AB T2N 4N1, Canada; antoine.dufour@ucalgary.ca; 3Department of Oral Biological and Medical Sciences and Centre for Blood Research, University of British Columbia, Vancouver, BC V6T 1Z4, Canada; chris.overall@ubc.ca

**Keywords:** Gelatinase A, Mmp2, zebrafish, sarcomere, myosin, proteostasis, phosphorylation, TAILS, secretion

## Abstract

Gelatinase A (Mmp2 in zebrafish) is a well-characterized effector of extracellular matrix remodeling, extracellular signaling, and along with other matrix metalloproteinases (MMPs) and extracellular proteases, it plays important roles in the establishment and maintenance of tissue architecture. Gelatinase A is also found moonlighting inside mammalian striated muscle cells, where it has been implicated in the pathology of ischemia-reperfusion injury. Gelatinase A has no known physiological function in muscle cells, and its localization within mammalian cells appears to be due to inefficient recognition of its N-terminal secretory signal. Here we show that Mmp2 is abundant within the skeletal muscle cells of zebrafish, where it localizes to the M-line of sarcomeres and degrades muscle myosin. The N-terminal secretory signal of zebrafish Mmp2 is also challenging to identify, and this is a conserved characteristic of gelatinase A orthologues, suggesting a selective pressure acting to prevent the efficient secretion of this protease. Furthermore, there are several strongly conserved phosphorylation sites within the catalytic domain of gelatinase A orthologues, some of which are phosphorylated in vivo, and which are known to regulate the activity of this protease. We conclude that gelatinase A likely participates in uncharacterized physiological functions within the striated muscle, possibly in the maintenance of sarcomere proteostasis, that are likely regulated by kinases and phosphatases present in the sarcomere.

## 1. Introduction

Matrix metalloproteinases (MMPs) are a complex family of about two dozen zinc-dependent proteases classically known for their roles in extracellular matrix (ECM) remodeling during development and disease, particularly in the contexts of inflammation and tumor biology [1,2,3,4,5,6,7]. They are broadly classified on the basis of their substrate specificities and structures into ‘Collagenases’ (MMPs 1, 8, 13, and 18), ‘Gelatinases’ (MMPs 2 and 9), ‘Metalloelastase’ (MMP12), ‘Stromelysins’ (MMPs 3, 10, and 11), ‘Matrilysins’ (MMPs 7 and 26), ‘Enamelysin’ (MMP20), ‘Epilysin’ (MMP28), ‘Membrane Type MMPs’ (MMPs 14, 15, 16, 17, 24, and 25), and ‘Other MMPs’ (MMPs 19, 21, 23, and 27) [3]. MMPs are generally localized to the extracellular space via type I secretion [8], regardless of whether they are secreted or membrane-type MMPs, with the exception of MMP23. Classical type I secretion via recognition of an N-terminal signal peptide targets other MMPs to the endoplasmic reticulum during translation, whereas MMP23 enters the ER/Golgi network post-translationally (i.e., it undergoes type II secretion), and is presented on the cell surface with its carboxyl end facing extracellularly [9]. MMPs are released into the extracellular environment or presented on the cell surface as inactive zymogens, and require the removal of an N-terminal propeptide (through the activity of pro-protein convertases or other proteases, including other MMPs) in order to become active [3,10]. Importantly, this irreversible post-translational proteolytic activation is emerging as one of, if not *the* most important level at which MMP activity is regulated, making the many reports focusing on changes in expression at the mRNA level difficult to interpret, as the biologically relevant activity is not well correlated with mRNA levels [11]. Novel approaches that focus on this post-translational activation (e.g., [12]) provide exciting opportunities to understand the regulation of MMP activity in vivo better. Once active, MMPs cleave a wide variety of extracellular matrix (ECM) and non-matrix proteins, including cell adhesion molecules, solute carriers, membrane receptors, and signaling molecules, and participate in a myriad of pathological and cell biological processes above and beyond matrix remodeling [3,6,13,14,15]. In addition to these well-established and undeniably important extracellular functions, many MMPs are also detected intracellularly in a variety of mammalian cell types [16,17,18]. They have been found in the cytosol [19,20,21,22], within the nucleus [20,23,24], and within mitochondria [19,22]. The mechanism(s) resulting in intracellular localization and the roles they play in these contexts remains poorly understood.

Gelatinase A (in humans the gelatinase A protein is called MMP-2, in mice it is referred to as MMP2, and in zebrafish as Mmp2; we have endeavored to be consistent with the naming conventions of the organisms in question, and have used ‘gelatinase A’ as the generic descriptor) is among the best-studied of the MMPs, and it is present nearly ubiquitously in embryonic and adult tissues of all vertebrates that have been examined. Surprisingly, mice deficient for MMP2 are viable and exhibit only subtle phenotypes (reviewed in [25]). However, anti-sense mediated knockdown of Mmp2 in zebrafish results in dramatic perturbations of embryonic development [26]. This is likely due to a combination of reduced redundancy between MMPs in zebrafish and their more rapid development providing less opportunity for compensatory mechanisms to mitigate the loss of Mmp2 activity [27]. Gelatinase A is among the MMPs found intracellularly [19,21,22,28], and it has been the focus of significant attention in the context of ischemia/reperfusion injury in cardiac muscle [29,30,31,32]. In human and murine myocytes, immunogold localization suggests it is concentrated in the sarcomeres at the Z-discs [19,22]. In human cells, MMP-2 protein accumulates intracellularly due to a poorly recognized N-terminal secretory signal; replacement of this sequence with a stronger signal sequence results in dramatically more efficient secretion, and N-terminal addition of the MMP-2 secretory signal to proteins otherwise efficiently targeted to the secretory pathway results in a dramatic reduction in the efficiency of this targeting [21].

Like research into their extracellular functions, investigations into intracellular functions of MMPs (including gelatinase A), have focused primarily on their pathological activities. In the context of mammalian cardiac muscle, ischemia/reperfusion events result in the production of reactive oxygen species (ROS), which can directly or indirectly modify the sulfhydryl group of the cysteine switch present in the autoinhibitory propeptide of gelatinase A, activating the protease [33]. Once activated, gelatinase A degrades several sarcomeric proteins, resulting in loss of contractility [31,34]. The upshot of this is that inhibition of gelatinase A activity is a promising avenue for mitigating the damage of ischemia/reperfusion injury in a clinical setting [30,31,32], but the question of why this potentially dangerous protease accumulates within the myocytes in the first place – i.e., what, if any, physiological functions does gelatinase A have in the sarcomere – remains unaddressed.

As well as cardiac muscle, gelatinase A has recently been detected in mammalian skeletal muscle [22,28]. Curiously, for an ostensibly extracellular enzyme, its proteolytic activity is subject to regulation by phosphorylation [35,36,37]. While extracellular kinases exist, and there are other examples of proteins with extracellular functions that are modulated by phosphorylation [38,39], the vast majority of kinases and phosphatases function intracellularly, and the sensitivity of mammalian gelatinase A to this type of regulation is surprising in the absence of any known intracellular function. It has been suggested that the inefficient secretion and susceptibility to regulation by phosphorylation of mammalian MMP2 may be ‘evolutionary spandrels’; quirks of this particular protease that have arisen as a result of unrelated adaptive changes that lack sufficient negative consequence to drive strong selection against them. If this were the case, we would not expect to find intracellular localization and/or phosphorylation of gelatinase A orthologues conserved across distantly related species. Here we begin to address these questions by examining the localization of Mmp2 in developing zebrafish embryos, and by analyzing the conservation of sequence features relating to the targeting of the protease to the type I secretory pathway and its phosphorylation by kinases.

We find that zebrafish Mmp2 is distinctly localized within the skeletal muscle cells of both embryos and adults, but at the M-lines rather than Z-discs, and only after sarcomerogenesis is complete. The N-terminal secretory signals of gelatinase A orthologues are consistently poorly recognized as such across a broad phylogenetic array of taxa, suggesting a selective pressure to maintain an intracellular pool of the protease. Furthermore, there are several strongly conserved phosphorylation sites suggesting the existence of an undiscovered intracellular system of kinases and phosphatases that regulate the activity of gelatinase A within the sarcomere. Finally, we observe that the protein most notably protected from proteolysis by the inhibition of MMP activity in zebrafish embryos is muscle myosin heavy chain, suggesting that sarcomeric proteins are genuine MMP substrates in vivo and that the Mmp2 we detect in the M-lines of embryonic skeletal muscle is likely degrading this component of thick filaments under normal physiological conditions. Taken together, this data suggests there is an ancient and conserved role for gelatinase A in the regulated turnover of sarcomeric proteins in vertebrates.

## 2. Experimental Section

### 2.1. Zebrafish Husbandry

Embryos were collected by natural spawning from adult wildtype (Tübingen) zebrafish maintained on a 14 h light: 10 h dark schedule in a flow-through system at between 26 °C and 28.5 °C, and staged according to [40]. Embryos were raised in standard Embryo Rearing Medium (ERM: 13 mM NaCl, 0.5 mM KCl, 0.02 mM Na_2_HPO_4_, 0.04 mM KH_2_PO_4_, 1.3 mM CaCl_2_, 1.0 mM MgSO_4_, and 4.2 mM NaHCO_3_, pH 7.4) at 28 °C, dechorionated manually using fine forceps, and either fixed at the desired stage in Dent’s fixative (80% methanol, 20% dimethyl sulfoxide (DMSO)) overnight at 4 °C, or homogenized as described below. Adult tissues were obtained by selecting healthy individuals (both male and female), sacrificing by MS-222 overdose, and dissection on ice. Dissected tissues were fixed in Dent’s fixative as above. All work with zebrafish was done with the approval and under the supervision of the University of New Brunswick’s Animal Care Committee (ACC Protocols 10013 (approved May 14, 2010), 11016 (approved May 16, 2011), 12013 (approved May 10, 2012), 13001, 13013 (May 13, 2013) and 14014 (approved May 7, 2014)), in accordance with the Canadian Council on Animal Care Guidelines.

### 2.2. Immunofluorescence and Cryosectioning

Samples were washed with PBSTx (0.1% Triton X-100 in Phosphate-Buffered Saline (PBS: 137 mM NaCl, 2.7 mM KCl, 20 mM phosphate pH 7.3)) five times for five minutes to remove fixative, blocked in blocking buffer (5% bovine serum albumin (BSA) in PBSTx) overnight at 4 °C, and incubated with primary antibodies; rabbit anti-zebrafish-Mmp2 (Anaspec catalog #55111, Freemont, CA, USA) and mouse anti-α-actinin (Sigma, catalog #A7811, Oakville, ON Canada), diluted (1:1000) in blocking buffer overnight at 4 °C. Samples were washed another five times for five minutes with PBSTx and then incubated with fluorescent conjugated secondary antibodies (Alexa-488 conjugated goat anti-rabbit IgG, as well as Alexa-633 conjugated goat anti-mouse IgG (Invitrogen, Carlsbad CA, USA) in samples processed for double labeling) diluted (1:1000) in blocking buffer overnight at 4 °C. After the final incubation, they were again washed with PBSTx, five times for five minutes each, and imaged using a Leica SP2 laser scanning confocal microscope (Leica, Wetzlar, Germany) with 20 × 0.7 NA water immersion and 63 × 1.4 NA oil immersion lenses.

For cryosectioning, 72 hours post-fertilization (hpf) zebrafish embryos were processed for double-labeling immunofluorescence as described above, washed in PBSTx, and embedded in 2.3 M sucrose dissolved in PBS overnight at 4 °C. The following day, the embedded embryos were frozen with liquid nitrogen and cut into 500 nm ultrathin sections using a Leica Ultracut T ultramicrotome. Sections were mounted on poly-l-Lysine coated glass slides and imaged as described above.

Images were assembled, and scale bars added using FIJI [41], and overlapping confocal stacks were projected stitched for the composite image shown in Figure 1A using the pairwise stitching plugin [42].

### 2.3. Immunoblotting

Embryos were collected at specified stages, dechorionated, and anesthetized in 0.4 mg/mL buffered tricaine before being homogenized in TRIzol (Thermo Fisher Scientific, Waltham, MA, USA) and stored at −20 °C. Homogenates were thawed on ice, centrifuged to remove insoluble debris, and protein separated from nucleic-acid containing fractions, according to the manufacturers directions. Protein pellets were solubilized in SDS-PAGE sample buffer (45 mM Tris pH 6.8, 10% glycerol, 1% SDS, 0.01% bromophenol blue, 50 mM DTT) and concentration was determined by BCA assays (Pierce, Rockford IL, USA). Three-hundred and fifty µg of protein from each stage was resolved on a 12% acrylamide gel, transferred to PVDF membrane, and the membrane incubated in blocking buffer (PBS with 0.1% Tween-20 (PBSTw) and 5% *w*/*v* skim milk powder) for 2 h at RT with gentle agitation. Membranes were probed with anti-zebrafish-Mmp2 (Anaspec, Freemont, CA, USA) diluted at 1:5000 in blocking buffer overnight at 4 °C. Unbound antibody was removed with three ten minute washes in PBSTw with gentle agitation at room temperature, and bands were visualized by incubating for 2 h at RT with HRP-conjugated goat anti-rabbit IgG (Invitrogen, Carlsbad, CA, USA) diluted 1:10000 in blocking buffer, followed by three ten minute washes in PBSTw and detection using Pierce^TM^ ECL Plus Western Blotting Substrate (ThermoFisher Scientific).

### 2.4. Analysis of Signal Peptides

Gene records for MMPs were retrieved from the NCBI Genbank database via keyword searches. The set of the longest protein sequences from each gene record were curated manually to include only sequences for which the gene symbol indicates an MMP. The sequence sets for each MMP, and the vitronectin control, were then aligned using MUltiple Sequence Comparison by Log-Expectation (MUSCLE 3.8.31 [43]), and the alignment was trimmed in AliView [44] to include only entries with highly conserved sequences in the first two dozen amino acids to avoid annotation errors in starting position or isoforms with divergent start positions. Sequences were removed from the analysis if they had gaps in the first fifteen amino acids, or if their start position was divergent from the consensus. The resulting set of sequences were processed using SignalP 4.1 [45] and mean S scores for each protein were graphed using R [46,47,48].

### 2.5. Phosphorylation Score Analysis

Phosphorylation scores for all orthologues of gelatinase A were predicted using NetPhos [49,50]. A multiple sequence alignment calculated using MUSCLE [43] was used to create a gap-map to translate the locations of possible phosphorylation sites (exported with the position in the amino acid sequence) from sequence coordinates into alignment coordinates and then back into the coordinates of a reference sequence (in this case the *Danio rerio* sequence). Sites were then restricted to those with scores higher than 0.65 and conservation in at least 97% of sequences (202 sequences out of 208). The line segment plot and domain map with disulfide bridges were drawn in R (R Core Team, 2017, https://www.r-project.org/), with aesthetics as used elsewhere for human MMP2 analysis [35].

### 2.6. Terminal Amine Isotopic Labeling of Substrates (TAILS)

TAILS identification of MMP substrates was performed as previously described [51]. Briefly, 2000 embryos were reared at 28 °C in ERM until 24 hpf, manually dechorionated with fine forceps, and then split into groups of 1000 and incubated at 28 °C with 3 µM phenanthroline (Sigma) or DMSO (vehicle) controls until 48 hpf and flash-frozen in liquid nitrogen. Proteomes were prepared as previously described, and their quality verified by SDS-PAGE. Proteomes were reduced and alkylated, and primary amines were labeled using 40 mM formaldehyde and 20 mM cyanoborohydride, using heavy (^13^CD_2_O and NaBD_3_CN) isotopically labeled reagents in the control reactions, and light (^12^CH_2_O and NaBH_3_CN) in the MMP-inhibited proteome, at 37 °C overnight. Labeled proteomes were precipitated with ice cold methanol, resolubilized in 50 mM HEPES pH 7.5, and digested with 5 µg of Trypsin Gold (Promega) overnight at 37 °C. Completion of trypsin digestion was verified by SDS-PAGE, and the peptides with newly generated amino-termini were removed by coupling to amine-reactive polymer, leaving only labeled N-terminal peptides from the original proteomes free in solution. Peptides were collected by filtration using 10 kD cut-off ultrafiltration (Amicon), pooled and acidified to pH 3 using formic acid, and bound to C_18_ stage tips before analysis by mass spectrometry using an LTQvelos Orbitrap as previously described [52]. Peptides were identified using MaxQuant (1.4.1.2) and the zebrafish UniProt (FASTA, 2014, *Danio rerio*) database. Peptides with a heavy:light ratio >3 were identified as amino termini significantly enriched in the control proteome, and therefore being representative of substrates protected by inhibition of MMPs.

## 3. Results

### 3.1. Mmp2 is Expressed after Gastrulation and Accumulates Intracellularly at the M-Lines of Skeletal Muscle Sarcomeres

Previous studies of *mmp2* expression in zebrafish embryos using in situ hybridization and RT-PCR suggested ubiquitous expression from early cleavage (1.5 h post-fertilization (hpf)) onwards [26], but no data on the distribution of the protein in zebrafish embryos has been published. Immunoblots of whole embryo homogenates probed with an antibody against zebrafish Mmp2 reveal little expression prior to the completion of gastrulation and the onset somitogenesis (~12 hpf), consistent with publicly available RNAseq data (http://www.ebi.ac.uk/gxa/experiments/E-ERAD-475). Both transcript and protein increase in abundance until about 72 hpf (Figure 1). Indirect immunofluorescence reveals broad, indistinct immunoreactivity starting during segmentation (data not shown). As the embryo develops, Mmp2 becomes increasingly abundant in the skeletal musculature, exhibiting apparently sarcomeric staining at high magnifications (Figure 1A and B). Strong immunoreactivity is also apparent associated with the posterior tip of the notochord, actinotrichia of the fins, neuromasts, mesenchymal cells in the head and eyes, and some axonal projections in the eyes and musculature. These axonal projections are not immunoreactive with the sensory neuron-specific antibody Zn-12 [53], suggesting they are likely a subset of motor-afferents (Appendix A).

Double immunofluorescence using an antibody against the Z-disc component α-actinin reveals Mmp2 immunoreactivity in distinct bands localized precisely between the α-actinin-positive Z-discs, corresponding with the M-line of the sarcomeres, in both embryonic and adult skeletal muscle (Figure 2). Consistent with M-line localization, Mmp2 immunoreactivity co-localizes with myosin heavy chain labeled with mAb F59 [54] (Appendix A). The dimensions of the confocal voxels in these micrographs have an axial resolution of 0.617 µm (calculated using https://svi.nl/NyquistCalculator), which is substantially less than the diameter of an adult myofibril (>10 µm), but uncomfortably close to the diameter of an embryonic myofibril (~1 µm). It is therefore theoretically possible that the Mmp2 immunoreactivity in the embryonic musculature could be superficial, rather than genuinely intracellular. To eliminate this possibility, we repeated the α-actinin/Mmp2 double immunofluorescence on ultrathin (500 nm) cryosections of skeletal muscle from 72 hpf embryos, thereby physically eliminating the possibility of superficial staining artifactually appearing between α-actinin immunoreactive Z-discs. Again, we observe robust Mmp2 immunoreactivity precisely between α-actinin positive Z-discs (Figure 2C). In all three preparations, the intensity of Mmp2 immunoreactivity correlates negatively with the intensity of α-actinin staining (correlation coefficients of −0.537, −0.496, and −0.649 in the confocal images of embryonic, adult, and cryosectioned embryonic muscle, respectively). This demonstrates unequivocally that Mmp2 protein accumulates at the M-lines of zebrafish skeletal muscle.

### 3.2. Mmp2 Accumulates at M-Lines Subsequent to Sarcomere Assembly

The skeletal musculature of vertebrates is derived from mesodermal somites, which form in zebrafish embryos starting around 11 hpf, with functionally contractile musculature arising around 18 hpf [40,55]. Since Mmp2 protein begins to accumulate prior to muscle cell differentiation and sarcomere formation, we wondered if the concentration of this protease in the sarcomeres begins before or after the assembly of the Z-discs from Z-bodies, during early sarcomereogenesis [56]. α-actinin/Mmp2 double immunofluorescence of the differentiating myofibrils in the trunk of 24 hpf embryos reveals the formation of nascent Z-discs in the periphery of early multi-nucleate myofibrils, but Mmp2 immunoreactivity remains roughly homogeneous within these cells at this stage, showing no strong correlation or anticorrelation with α-actinin immunoreactivity (correlation coefficient of 0.334) (Figure 3), suggesting that whatever role Mmp2 plays, its accumulation does not precede sarcomere formation.

### 3.3. Gelatinase A Orthologues have poorly Recognized Signal Sequences

The apparent abundance of Mmp2 within the skeletal muscle cells of the zebrafish embryos and adults led us to wonder if the zebrafish Mmp2 protein has an inefficiently recognized secretory signal peptide, as is the case in the human MMP-2 [21]. SignalP 4.1 is a software neural network trained to recognize class I secretory signals in eukaryotic proteins using known secreted proteins [45]. It is used routinely to identify secreted proteins in zebrafish and other vertebrates [57]. SignalP outputs C, S, Y, and D scores, representing predicted signal peptide cleavage sites, overall ‘signal peptideness’, a combined cleavage site prediction, and a final type I secretory protein ‘discrimination score’, respectively [45]. The zebrafish Mmp2 protein has an N-terminal secretory signal, but like the human MMP-2 protein, this sequence is barely recognizable by SignalP (zebrafish Mmp2 S_mean_ = 0.795; human MMP-2 S_mean_ = 0.845; typical secreted proteins have S_mean_ scores > 0.9), suggesting that, as is the case in mammals, a significant proportion of the Mmp2 protein produced in zebrafish cells escapes recognition by the signal recognition particle and is retained in the cytosol after translation. Given that the secretory signal is removed from the N-terminus of the protein as it is translocated through the Sec61 complex [58], the sequence of this part of the protein cannot affect the folding or activity of the rest of the molecule once it enters the secretory pathway. So it is difficult to imagine any selective pressures that would prevent this sequence converging on an efficiently recognized secretory signal unless there are advantageous physiological processes that require gelatinase A within the cell (where this N-terminal sequence would be retained). We tested this hypothesis by comparing the S scores of 208 gelatinase A orthologues to those of orthologues of an extracellular matrix protein that is efficiently secreted (vitronectin) and those of all other secreted MMPs, including orthologues of MMP23, which undergoes type II secretion and therefore lacks a conserved N-terminal secretory signal (Figure 4). As expected, orthologues of vitronectin have consistently well-recognized signal sequences, and orthologues of MMP23 have N-termini without recognizable signal sequences. Interestingly, while almost all other secreted MMPs have easily recognizable signal sequences, orthologues of gelatinase A (and MMP21) have consistently less recognizable N-terminal secretory signals. This suggests that there are selective pressures acting against efficient secretion of gelatinase A and MMP21.

### 3.4. Gelatinase A has Conserved Phosphorylation Sites that Regulate its Enzymatic Activity

Consistent with it having unrecognized intracellular functions, the proteolytic activity of gelatinase A is regulated by phosphorylation [35]. NetPhos 4.1 predicts both generic and kinase-specific phosphorylation sites in eukaryotic proteins [49,50], and several of the sites it identifies in human MMP-2 have been empirically verified [37]. Of the gelatinase A sequences currently known, there are five predicted phosphorylation sites that are conserved in all 208 orthologues (Figure 5). Two of these are sites that have been empirically verified as being phosphorylated in vivo, and all of them occur within the catalytic domain of the protein. There are five more sites that are conserved in 99% of sequences, and another three that are conserved in 97% of gelatinase A sequences. The extremely strong conservation of these known and predicted phosphorylation sites suggests they are under functional constraint, implying the existence of regulatory kinases and phosphatases that modulate the activity of this protease in an intracellular context.

### 3.5. Myosin is a Target of Metalloproteinases in Zebrafish Embryos in vivo

A longstanding problem in MMP biology has been the identification of genuine substrates in vivo [59,60,61,62]. Recently, terminal amino isotopic labeling of substrates (TAILS) has been used to quantitatively compare the abundance of novel N-termini between proteomes of samples collected under conditions with and without the activity of (a) specific protease(s), in order to generate ‘degradomes’ that allow the identification of proteins that are genuine targets of the protease(s) in question [51]. We used TAILS to compare the degradomes of 48 hpf zebrafish embryos in the presence or absence of the broad-spectrum MMP inhibitor phenanthroline [63]. We were able to identify 321 peptides using the zebrafish UniProt database, 49 of which exhibited heavy/light ratios greater than 3, indicating they are likely targets of proteases inhibited by phenanthroline (Data S1). A disproportionately high 24% (12/49) of these are fragments of myosin, four of which have isoleucine, leucine, or valine in the P1′ position characteristic of gelatinase A cleavage sites [64] (Table 1). Thus, it appears that myosin is a genuine physiological target of MMPs in vivo, and it is likely that Mmp2 contributes at least some if not all, of the relevant proteolytic activity targeting this primary component of the thick sarcomeric filaments.

## 4. Discussion

Gelatinase A is well known for its roles in ECM remodeling during development and wound healing, and as a central player in a wide variety of pathologies such as tumor metastasis and fibrosis of various tissues [65]. It also has key roles in complement activation [66] and in the regulation of chemokine and cytokine signaling in vivo [6,7]. In addition to these extracellular roles, Gelatinase A functions as an intracellular enzyme [16], but the mechanism(s) by which it becomes resident intracellularly and its function(s) in this context remain poorly understood. Why this potentially dangerous protease is not secreted more efficiently, thereby protecting cells from its inappropriate intracellular activation, is a mystery. Mutations in the sequence encoding the N-terminal secretory signal cannot affect the function of the protein once it enters the secretory pathway as the signal sequence is removed during translation [8,57], so one would expect the signal sequence of any protein that functions exclusively extracellularly to converge on one that is easily recognized by the signal recognition particle. This appears to be true for most secreted proteins, including most MMPs, but not for orthologues of gelatinase A.

The N-terminal secretory signal of human MMP-2 is inefficiently recognized, and secretion of MMP-2 can be dramatically improved by mutating this sequence such that it is better recognized by the signal recognition particle [21]. This is not an idiosyncrasy of the human protein; we observe that orthologues of gelatinase A across a broad phylogenetic array of taxa have N-terminal secretory signals that are consistently difficult to recognize, suggesting that there is a selective pressure that maintains this curious inefficiency, effectively splitting the production of gelatinase A protein between the outside and the inside of cells expressing this gene. Consistent with this, in the zebrafish Mmp2 protein is present within the myocytes of embryonic and adult skeletal muscle, demonstrating that this is a general feature of striated muscle cells and not a peculiarity of mammalian cardiac myocytes or mammalian cells in general. It is difficult to envision a scenario in which it would be advantageous to maintain a portion of any protein, let alone a potentially dangerous protease, within the cell if there were no adaptive functions for that molecule to fulfill within the cell, especially when mutations that increase the efficiency of secretion do not require changes to the sequence of the mature protein.

We detect Mmp2 at the M-lines of sarcomeres in skeletal muscle, and our TAILS data demonstrates that the myosin of thick filaments found in the M-band is degraded by MMPs—likely including Mmp2—under normal physiological conditions. This inconsistency with previous reports localizing mammalian MMP2 to Z-discs [19,22], may represent a genuine difference between cardiac and skeletal myocytes, or between mammalian and teleost myocytes, or both. Alternatively, this may represent an aspect of the functional regulation of gelatinase A activity; it may shuttle between these locations, similarly to the shuttling between the Z-disc and A-band exhibited by the myosin chaperones Unc45b and Hsp90 [67]. It is tempting to speculate that such changes in the localization of gelatinase A might be regulated by changes in its phosphorylation status.

Gelatinase A purified from mammalian cells is definitely phosphorylated, and while we may speculate on how its phosphorylation status may alter its localization, there is no doubt that phosphorylation profoundly alters its enzymatic activity [35,36,37]. This is likely true of zebrafish Mmp2, and the conservation of verified and putative phosphorylation sites suggests it is likely true of all gelatinase A orthologues. While ATP is generally not abundant extracellularly, extracellular kinases do exist and tissue inhibitor of matrix metalloproteinases 2 (TIMP-2) is phosphorylated in vivo [38,68,69]. So we cannot rule out the possibility that the phosphorylation of gelatinase A functions to regulate its extracellular activity. However, the most abundant extracellular kinases phosphorylate serine within SxE motifs [69], and none of the putative or known phosphorylation sites in gelatinase A match this pattern, nor are there known examples of other extracellular proteases that are regulated by phosphorylation.

The M-band of the sarcomere is rich in kinases and phosphatases that are known to function in regulating the turnover of sarcomeric proteins [70], although the specific mechanisms they regulate remain somewhat obscure. This suggests to us that gelatinase A may play a role in maintaining the proteostasis of the sarcomere, possibly contributing to the complex system mediating recycling of surplus and damaged components of the thick filament during muscle cell development and in response to exhaustive exercise [71,72,73]. The phosphorylation status of gelatinase A may therefore indicate the physiological status of the myocyte, and the zebrafish is emerging as an excellent model system for investigating these sorts of questions regarding muscle cell development and physiology [74,75].

The extracellular functions of gelatinase A and other MMPs remain important research questions, but we must not neglect the intracellular functions of these proteins or consider them only in pathological contexts [18]. Understanding the normal physiological mechanisms in which these molecules participate, their molecular regulation, their development, and their evolutionary origins will undoubtedly provide insights that are valuable in treating the pathologies that arise as a result of their misregulation. There is clearly selective pressure on orthologues of gelatinase A and MMP21 to maintain an intracellular pool of these proteases; Mmp2 is consistently present in the sarcomeres of striated muscle from fish to mammals, and sarcomeric proteins are predominant in the zebrafish MMP degradome in vivo. Taken together with the conservation of putative and empirically verified phosphorylation sites that regulate this proteolytic activity (and possibly localization) of this protease, we conclude that gelatinase A likely plays an important unrecognized role in the physiology of striated muscle that deserves more thorough investigation.

## Figures and Tables

**Figure 1 biomedicines-07-00093-f001:**
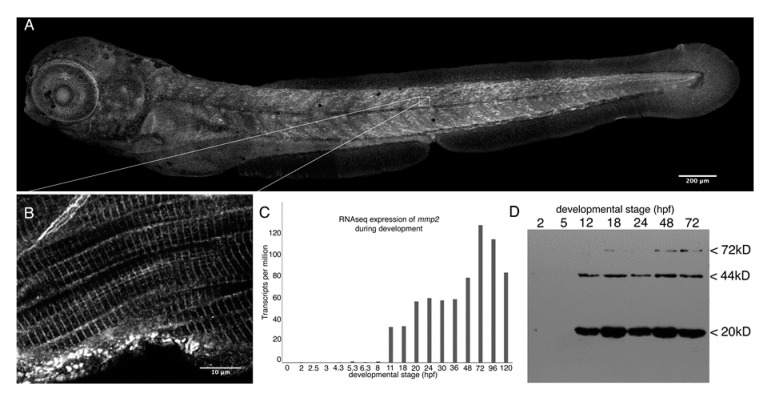
Mmp2 is expressed ubiquitously from early development and accumulates in a striated pattern within the skeletal muscle. (**A**) Composite confocal projections of a 72 hpf embryo stained with anti-Mmp2 exhibiting labeling throughout the embryo with notable accumulation in the skeletal muscle (scale bar = 200 µm). (**B**) High magnification view of a single confocal section through the trunk musculature (indicated by the inset) showing strong labeling of the myotome boundary (upper left corner), and striated staining in myofibrils (scale bar = 10 µm). (**C**) RNASeq data showing absolute abundance of *mmp2* transcripts in embryos from fertilization to five days post-fertilization (dpf). (**D**) Immunoblot of whole embryo homogenates (350 µg per lane) made from 2 hpf (cleavage), 5 hpf (50% epiboly), 12 hpf (early somitogenesis), 18 hpf (late somitogenesis), 24 hpf (prim-5), 48 hpf (long pec), and 72 hpf (protruding mouth) embryos probed with anti-Mmp2. Immunoreactivity is detected in embryos after 12 h of development, with bands at the expected mobility for full-length Mmp2 (72 kD) and stronger bands at 44 and 20 kD, which combine to give the expected size for activated Mmp2 (64 kD).

**Figure 2 biomedicines-07-00093-f002:**
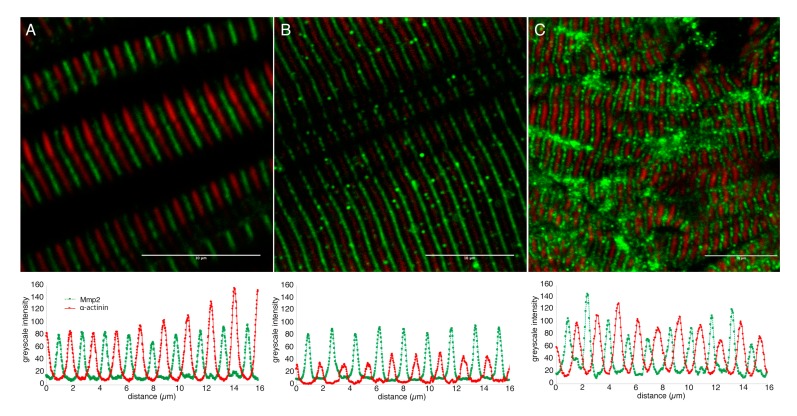
Mmp2 is localized between Z-discs in sarcomeres of embryonic and adult muscle. Confocal micrographs of skeletal muscle from 72 hpf embryos (**A**) or adults (**B**), and 500 nm thick cryosection of 72 hpf skeletal muscle (**C**) stained with anti-α-actinin (**red**) and anti-Mmp2 (**green**). Greyscale intensity profiles of both channels along a line drawn perpendicular to the sarcomeres are shown below each micrograph. Mmp2 immunoreactivity occurs at regularly spaced intervals precisely between the α-actinin-labeled Z-discs. Scale bars = 10 µm.

**Figure 3 biomedicines-07-00093-f003:**
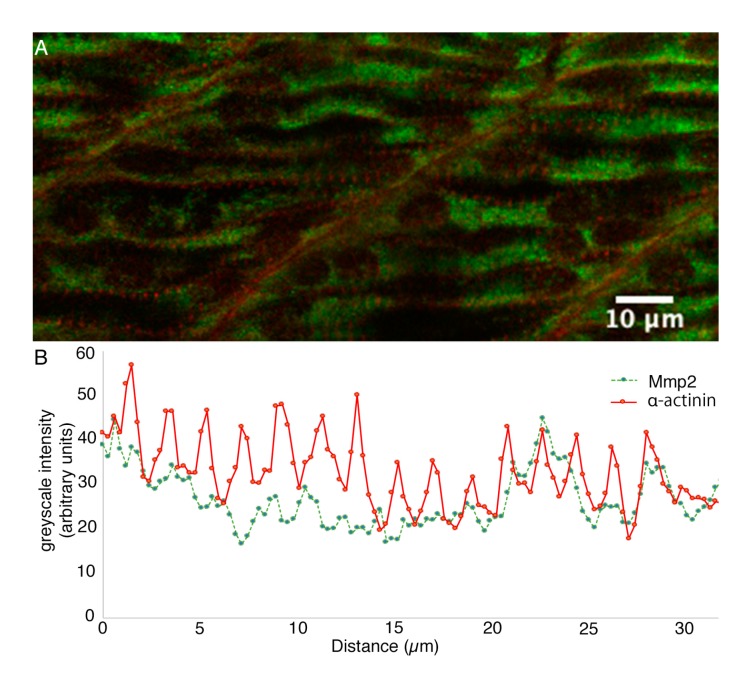
Sarcomeric Mmp2 begins to accumulate subsequent to the assembly of Z-disks. (**A**) Confocal section through the trunk musculature of a 24 hpf embryo posterior to the yolk extension, at the position at which myofibrils are differentiating. (**B**) Sarcomeric α-actinin (**red**) is beginning to become apparent as Z-bodies in the periphery of differentiating myocytes, but Mmp2 immunoreactivity (**green**) remains roughly homogeneously distributed. Greyscale intensity profiles of both channels along a line drawn through the periphery of a differentiating myocyte are shown below the micrograph. Scale bar = 10 µm.

**Figure 4 biomedicines-07-00093-f004:**
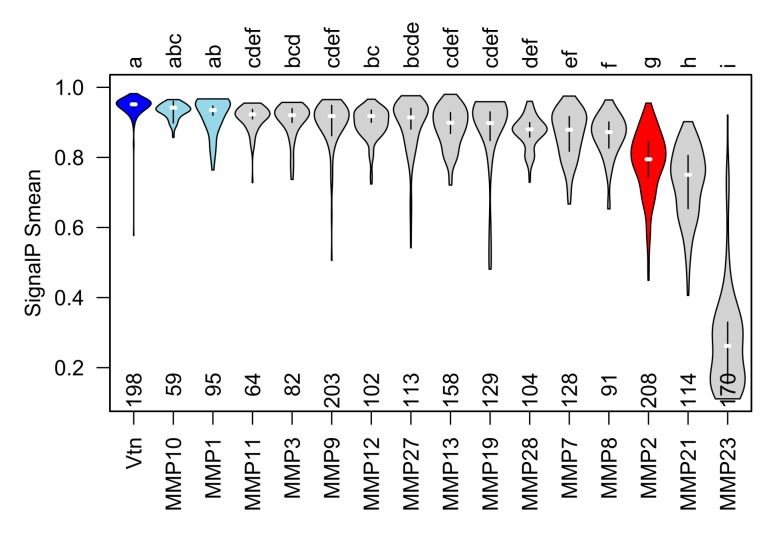
The secretory signal peptide of gelatinase A orthologues is consistently and significantly less likely to be recognized than that of most other type-I secreted proteins. Violin plots of mean ‘S’ scores of the N-terminal secretory signals from orthologues of vitronectin (Vtn) and all secreted MMPs. MMPs with mean S score statistically indistinguishable from vitronectin are shown in blue. Mean S scores for orthologues of gelatinase A (red) are significantly lower than those of vitronectin and other secreted MMPs apart from MMP21 and MMP23, which undergoes type II secretion and therefore, does not have an N-terminal secretory signal. Statistically, indistinguishable groups are indicated with letters at the top of the plot, and the number of orthologues of each protein analyzed is indicated along the *x*-axis.

**Figure 5 biomedicines-07-00093-f005:**
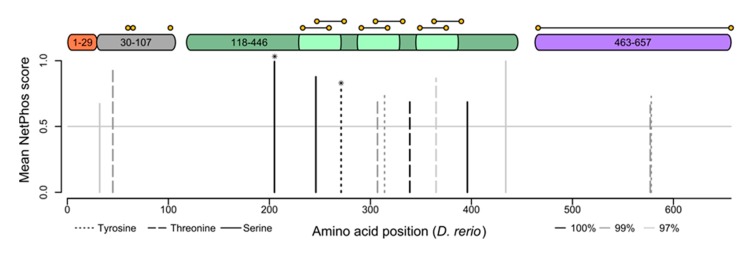
Gelatinase A orthologues have highly conserved phosphorylation sites. Putative serine (solid lines), threonine (dashed lines), and tyrosine (dotted lines) phosphorylation sites conserved in 100% (**black**), 99% (**dark grey**), or 97% (**light grey**) of gelatinase A orthologues are shown with respect to a structural schematic of the gelatinase A protein, illustrating the signal sequence (1–29 (**orange**)), propeptide (30–107 (**grey**)), catalytic domain (118–446 (**green**)) with fibronectin-like repeats (light green), and hemopexin-like domain (463–657 (**purple**)). Cysteines are indicated with yellow spots, connected by horizontal lines if they are predicted to participate in intramolecular disulfide bonds. Conserved residues that have been empirically demonstrated to be phosphorylated in vivo in the human protein are indicated with asterisks.

**Table 1 biomedicines-07-00093-t001:** Myosin peptides represent 24% of significant hits in TAILS after inhibition of metalloproteinases in 48 hpf zebrafish embryos. (Bold indicates P1′ residues typical of Gelatinase A substrates).

Myosin Peptide	P1′ Residue	Normalized H/L Ratio
DLEESTLQHEATAAALR	Asp	8.3856
IEELEEELEAER	**Ile**	7.6228
ELETEIEAEQR	Glu	6.6756
ADLSRELEEISER	Ala	4.6963
VRELESEVEAEQR	**Val**	4.4234
TLEDQLSEIKSKNDENLR	Thr	3.4527
VQLELNQVKSEIDR	**Val**	3.4453
LEDEEEINAELTAKKR	**Leu**	3.3276
ELESEVEAEQR	Glu	3.1041
ADIAESQVNKLR	Ala	3.0920
EQFEEEQEAKAELQR	Glu	3.0497
QLEEKEALVSQLTR	Gln	3.0294

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
