# Peer review of "Intracellular Localization in Zebrafish Muscle and Conserved Sequence Features Suggest Roles for Gelatinase A Moonlighting in Sarcomere Maintenance"

_biomedicines, 2019, doi:10.3390/biomedicines7040093_

Round 1
Reviewer 1 Report
The authors have addressed my concerns.
Please note that on page 7, line 10 should refer to Figure 2C, not 1C.
Author Response
The reference to figure 1C has been corrected to 2C. However, this was on line 42 of page 6 in our copy, which suggests the version seen by the reviewers are different than what the Journal has returned to us.
Reviewer 2 Report
Previous comments have been well addressed, minor comments:
Moonlighting is not the best language in title but will leave at authors discretion Please add more labels to figure 1D what does each row represent and the bottom part of figure is lost maybe in submission. Figure 4 needs aligning properly might be an issue in submitting to journal
Author Response
We understand the Reviewer's discomfort with the 'moonlighting' colloquialism, however this terminology has become established as an informal way of identifying research relevant to the intracellular functions of MMPs in the literature. We hope that the use of this shorthand will help prospective readers interested in this topic find and recognize this report as being relevant.
We are not sure what additional labeling the western blot shown in figure 1D could be added; this is typically how ontogeny westerns are illustrated.
Similarly, we don't see any alignment issue with figure 4.
We surmise that the PDF copy of the manuscript provided to the Reviewer may not have had complete figures?
Reviewer 3 Report
I have no further comments.
Author Response
We would like to thank the Reviewer for their constructive comments on the first submission.
This manuscript is a resubmission of an earlier submission. The following is a list of the peer review reports and author responses from that submission.
Round 1
Reviewer 1 Report
Fallata and colleagues present a straightforward study on the zebrafish Mmp2, a protein which presents some novelties among metalloproteases. They make a salient point about the need to study this gene product at the protein level, given that phosphorylation is a common mediator of Mmp activity in general. Mmp2 protein accumulates after gastrulation. Intriguingly, the subcellular localization appears to be primarily intracellular, where it localizes to the M lines of zebrafish trunk musculature. The findings that M line accumulation occurs after Z body formation, and that myosin heavy chain is a Mmp target (possibly a Mmp2 target) are interesting considering what possible functions Mmp2 might play intracellularly. The analysis of peptide signal and phosphorylation site conservation over 208 orthologues and several Mmp genes are consistent with the putative intracellular localization of the protein.
The limitations of this study are that experimental evidence is not included to demonstrate whether the conserved phosphorylation sites are actually used in zebrafish embryonic trunk, or whether myosin heavy chain is actually an Mmp2 target. While the data do support a primarily intracellular localization, this study does not determine what the function of Mmp2 actually is, in that context. That being said, the report in its current form does, to my mind, sufficiently represent an advance to the field and likely will stimulate further studies with more functional focus. The manuscript is quite well written. Especially, I find the speculation that Mmp2 might shuttle between M line and Z bands is fascinating.
Concerns
The text in Figure 1C is so small it is completely unreadable. Font size and text resolution need to be fixed. In Figure 3, with the current magnification it is difficult to perceive the regularly spaced alpha actinin localization. If the current figure is enlarged 400%, it is much easier to detect the linear z body arrangements and cell nuclei. On page 11, line 21 – C.M.O. is supported by ??
Author Response
- With respect to the limitations raised by this reviewer - i.e that we present no experimental data to demonstrate whether the conserved phosphorylation sites are actually used in zebrafish skeletal muscle, nor do we provide data to demonstrate an actual function for the intracellular Mmp2 - the Reviewer is entirely correct. We have struggled to address these limitations with very limited success, and decided that it would be valuable to publish the data we have while we pursue these more challenging experimental issues further. We have been able to purify small amounts of Mmp2 protein from adult zebrafish skeletal muscle using gelatin affinity chromatography, and we looked for evidence of phosphorylation using ProQ-Diamond staining as well as by western blotting with antibodies against phospho-serine, phospho-threonine and phospho-tyrosine. While there was some hint of phosphorylation in some of these assays, we were unable to reproducibly generate data that was of publication quality due to the limited amount of protein we are able to obtain using this approach (despite dissecting over a hundred adult zebrafish per attempt we were able to purify only barely enough Mmp2 to detect on a gel). We hope to pursue this, and the possibility that Mmp2 localization within the sarcomere is dynamic, in future investigations using larger animals (salmonids), but that is beyond the scope of the current manuscript.
- Text in figure 1C has been increased in size to improve legibility.
- Figure 3 has been merged into Figure 2 (as also suggested by another Reviewer) and we have increased the scale of the image to address this issue and make it more similar to the other two panels. We have also increased the scale of the micrograph from the old Figure 4 (new Figure 3) to improve visibility (this may have been what the Reviewer meant), and have added some quantitative analysis of these images to both figures.
- The grant support information for C.M.O. has been added.
Reviewer 2 Report
Dear Editor
This manuscript deals with an area of extracellular matrix biology which is gaining increasing attention. The role of MMP2 is a well established remodelling molecule in this manuscript the author investigates gelatinase a as also being involved in proteostasis of sarcomere. The author provides mainly a very descriptive and little to no mechanistic insight is presented, however the data is generally of reasonable quality and for this reason subject to major revisions I would like to suggest this manuscript is published. The manuscript is for the most part well written.The area of sarcomere proteostasis is growing and this would be a paper of interest to muscle biologists generally.
Major Revisions:
Fig 1 needs improving through out. 1a does not show anything useful, 1b is good but need time series to match western blot times, 1c needs better axis labels questionable if its previously published data if it is better in supplementary data will leave to authors discretion. western blot needs loading control looks unevenly loaded maybe use beta tubulin from abcam it is zebrafish specific. Fig 2 make scale bars same width, merge with fig 3 and 4, needs some quantification. Discussion needs to be less of a reiteration of results. Perhaps uses the word physiology when the author really means biochemistry.
Minor Comments:
Page 1:
Line 12 ", and" not correct punctuation Line 35 Define N terminal targets better Line 38 "facing out" consider extra cellular facing Line 41 transitionally activated is very vague define better
Page 2:
Line 7 "mechanism(s)" is non committal language line 37 "spandrels" questionable language choice
Page 3:
line 3 "bona fide" not scientific language
Author Response
- We have made some changes to figure 1 (increased font size in 1C to improve legibility), but we disagree with this reviewer's assessment that 1a is not useful. There is no published data regarding the distribution of Mmp2 protein in zebrafish (or any other non-mammalian vertebrate), and we contend that this high-resolution image of the immunostaining pattern in the whole 72 hpf embryo provides both valuable context for the rest of the data, as well as a record of other aspects of Mmp2 localization that we mention briefly but did not pursue in this study (e.g. notochord, neuromasts, etc.). Particularly given that this antibody is no longer sold, we felt that making this data available for future researchers would be of value.
That being said, we agree that the purpose of this manuscript is not to provide an extensive archive of Mmp2 immunostaining patterns. So with respect to a time series of whole mount immunostains to match the western blot, while we have some of this data and could include it (although scales are different an it would be very difficult to assemble into a coherent figure), we feel it would distract from the main thrust of this study; we provide a supplemental figure relating to the apparent neural localization, and more temporal resolution with respect to when the Mmp2 becomes sarcomerically localized in a subsequent figure, so we do not feel that additional micrographs illustrating the non-sarcomeric localization of this protease would add to this manuscript.
Finally, with respect to the lack of loading controls on the western blot in Figure 1, we understand and agree in principle with the Reviewer's concern. However, loading controls are not easy to implement in western blots of zebrafish embryos of these stages; early embryos have only a few cells and a large yolk, whereas later stages have many thousands of cells and less yolk. It is therefore effectively impossible to find a protein that does not change with respect to its relative abundance over the course of early development; typical "house keeping genes" like beta tubulin or GAPDH undergo dramatic changes in relative abundance over the course of the first few hours of development, and therefore cannot be used to normalize signal intensities on western blots of early stages. Instead, we have used identical total protein loading (350 µg) in each lane, as described in the methods section, and have added this to the figure legend as well. This approach of using identical protein loads is common and widely accepted for ontogeny blots of zebrafish embryos (for examples, see Henry et al. (2002). Roles for zebrafish focal adhesion kinase in notochord and somite morphogenesis. Developmental Biology, 240(2), 474–487. http://doi.org/10.1006/dbio.2001.0467; Crawford et al. (2003). Activity and distribution of paxillin, focal adhesion kinase, and cadherin indicate cooperative roles during zebrafish morphogenesis. Molecular Biology of the Cell, 14(8), 3065–3081. http://doi.org/10.1091/mbc.E02-08-0537; Manning et al. (2004). unc-119 homolog required for normal development of the zebrafish nervous system. Genesis, 40(4), 223–230. http://doi.org/10.1002/gene.20089; Wyatt et al. (2009). The Zebrafish Embryo: A Powerful Model System for Investigating Matrix Remodeling. Zebrafish, 6(4), 347–354. http://doi.org/10.1089/zeb.2009.0609; Vass et al.(2013). Perturbation of invadolysin disrupts cell migration in zebrafish (Danio rerio). Experimental Cell Research, 319(8), 1198–1212. http://doi.org/10.1016/j.yexcr.2013.02.005; De Angelis et al. (2017). Tmem2 Regulates Embryonic Vegf Signaling by Controlling Hyaluronic Acid Turnover. Developmental Cell, 40(2), 123–136. http://doi.org/10.1016/j.devcel.2016.12.017). Furthermore, as mentioned above, this antibody is no longer available, so even if we could find a suitable "loading control", we can not go back and repeat this experiment.
- We agree with the Reviewer that old Figures 2 and 3 were essentially making the same point. We have merged these figures into a new Figure 2, and have also adjusted the depiction and position of the scale bars to allow easy comparison. We have also added some quantitative image analysis to the new Figure 2 in order to make comparisons with the new Figure 3 more clear. These changes are also reflected in the text.
- The reviewer suggests that figure 4 "needs some quantification." We are not certain what is meant here, but we have done some quantitative image analysis of the greyscale pixel values for the green (Mmp2) and red (alpha actinin) channels (as well as correlation analysis) to support our conclusion that Mmp2 does not become sarcomerically organized until after Z-disc assembly. This has been added to the new Figure 3, and the discussion of this result. We have also expanded the scale of the micrograph to make the data more easily visible as suggested by another reviewer.
- With respect to the minor suggested edits to the text, we have corrected these where necessary. However, we have retained the use of "spandrels" as we are using it in Gould & Lewontin's sense of "an explicit term for features arising as byproducts, rather than adaptations, whatever their subsequent exaptive utility..." (Gould, 1997)
Reviewer 3 Report
The manuscript presents a very well written and interesting study about MMP2 in zebrafish and its unknown functions. The authors discuss that in a very good manner.
I recommend publishing this manuscript after minor revision since it presents an interesting view on proteases and possible functions.
Introduction:
Page 1 Line 31-33: The authors could shortly introduce the MMP protein family, e.g. number of members or if they are classified into subfamilies Sometimes it would be helpful to state in which species the cited results were obtained, e.g. page 2 line 10/11. Are there studies present with loss or gain of function mutations introduced in MMP2? If yes, what did they show? It might be helpful for the reader to introduce milestones in zebrafish muscle development and important time points of MMP2 expression in order to better understand the chosen time points for analyses.
Experimental Section:
Page 3 Line 12: “Embryos were raiseD…” – spelling error Page 3 Line 17: “disSection” – spelling error Page 3 Line 43: “Embryos were collected aT…” – spelling error Page 3 Line 44: “…tricaIne…” – spelling error Page 3 Line 47: Composition of the sample buffer? Page 4 Line 5: “HRP-conjugated goat anti-rabbit …” The word antibody seems to be missing. Page 4 line 6: Please give the full name of ECL-Plus Page 4 line 33: “previously describe” – where exactly? General: please provide the full company name, e.g. not only “Thermo”
Results:
Figure 1: A/B: Can you provide this as colored pictures? C: Labeling is too small. D: Please show a loading control. Figure 5: Can MMP2 be included here?
Acknowledgments:
Page 11 line 21: “C.M.O is supported by…” – please provide missing information.
Author Response
- We have expanded somewhat in the Introduction of MMPs to provide more background on the protein family and their classifications.
- The Reviewer raises an important point regarding mentioning the species in which research has been done; the vast majority of research on MMPs has utilized mammalian cells or murine model systems, and this mammalian bias prevents making important evolutionary inferences. We have adjusted the text to clarify and emphasize this point.
- We have added some text (and references) regarding the phenotypes of MMP2 knockouts in mice and Mmp2 knockdowns in zebrafish (and the important differences). Again, because of the historical bias towards using mammalian models, significant insights may be had by investigating non-mammalian models.
- We have added some text (and references) regarding the ontogeny of skeletal musculature in zebrafish, and its relationship to the timing of Mmp2 expression.
- The typos identified by the Reviewer (as well as a few others) have been corrected.
- The composition of the SDS-PAGE sample buffer used has been added
- Reagent and supplier names have been expanded
- The TAILS technique was previously described in the reference cited (now reference 51).
- We have used greyscale images for the single-channel confocal data in figure 1 because the human visual system can distinguish more shades of grey than of any colour (i.e. rod cells have a greater dynamic range than cone cells); it is therefore easier for the viewer to distinguish subtle details in the images if they are presented as greyscale than if they are presented in colour.
- labeling in figure 1C has been improved.
- With respect to the loading control for the western blot shown in 1D, as discussed in the response to the previous reviewer, due to the changing relative abundance of all proteins during early zebrafish development, there is no good loading control we could use for this blot, so we have used identical total protein loads (350 µg per lane) as is commonly done for ontogeny westerns of early zebrafish embryos (see references cited in response to item 1 for Reviewer 2).
- Figure 5 schematically illustrates the structure of Gelatinase A, the position, type and conservation of phosphorylation sites, with amino acid numbering based on the zebrafish Mmp2 sequence. So human MMP-2 and mouse MMP2 are included in the analysis.
- The grant support information for C.M.O. has been added.
Finally, we have added a few references and made a number of minor adjustments to the text and title to better integrate our results with the relevant literature. These changes will increase the likelihood that researchers interested in intracellular MMPs will find and read this report.